# Decision-Making Process in Female Genital Mutilation: A Systematic Review

**DOI:** 10.3390/ijerph17103362

**Published:** 2020-05-12

**Authors:** Angi Alradie-Mohamed, Russell Kabir, S.M. Yasir Arafat

**Affiliations:** 1School of Allied Health, Anglia Ruskin University, Chelmsford CM1 1SQ, UK; angi.alradie-mohamed@student.anglia.ac.uk; 2Department of Psychiatry, Enam Medical College and Hospital, Dhaka 1340, Bangladesh; arafatdmc62@gmail.com

**Keywords:** female genital cutting, female genital mutilation, female circumcision, decision-making process, decision-maker, attitude

## Abstract

Female genital mutilation/cutting “FGM/C” is a deep-rooted damaging practice. Despite the growing efforts to end this practice, the current trends of its decline are not enough to overcome the population’s underlying growth. The aim of this research is to investigate the FGM/C household decision-making process and identify the main household decision-makers. A review of peer-reviewed articles was conducted by searching PubMed, JSTOR, Ovid MEDLINE, Ovid EMBASE, EBSCO, and CINAHL Plus via systematic search using keywords. The found publications were screen using inclusion and exclusion criteria in line with Preferred Reporting Items for Systematic Reviews and Meta-Analyses (PRISMA) guidelines. After critical appraisal, seventeen articles were included in this review. The data extracted from the articles regarding FGM/C household-decision making process and decision-makers were analyzed using narrative analysis. FGM/C decision-making process varies from a region to another; however, it generally involves more than one individual, and each one has different power over the decision. Fathers, mothers, and grandmothers are the main decision-makers. It was shown from this review that opening the dialogue regarding FGM/C between sexes may lead to a productive decision-making process. The participation of fathers in the decision-making may free the mothers from the social-pressure and responsibility of carrying on traditions and create a more favorable environment to stop FGM/C practice.

## 1. Introduction

Female genital mutilation or female circumcision/cutting (FGM/C) is one of the most ancient deep-rooted damaging practices worldwide. It is defined by the World Health Organization (WHO) as “all procedures involving partial or total removal of the external female genitalia or other injuries to the female genital organs for non-medical reasons” [1]. It causes lifelong suffering for women and girls and creates health inequality without having any health benefits, by leading to life-lasting physical, mental and sexual problems among females who are exposed to the practice (severe pains, bleeding, death, kidney failure, infertility, difficulties during childbirth, fetal distress, death of the newborn, and/or maternal death) [2].The psychological consequences of FGM/C include behavior disturbances and loss of trust among girls and long-term consequences among women (depression, anxiety, feeling of incompleteness, and the inability to express their fears) [3].

The WHO and the United Nations Children’s Fund (UNICEF) (1997) [3], classify it to four major types: “clitoridectomy” or Type I which involves the excision of the clitoral prepuce “the hood” or/and excision of all/ part of the clitoris, Type II is the excision of the clitoris with excision of part or all of the labia minora, “infibulation” or Type III involves excision of part or all of the external genitalia and the sewing in or narrowing of the vaginal opening, Type III is the most invasive and harmful type among the others, while Type IV is called unclassified which includes any other harmful procedures or injures to the genitalia that falls under FGM/C definition. 

Although FGM/C is a human rights violation, in 2016, it was estimated that at least 200 million females in thirty countries were victimized by it [4]. Furthermore, the WHO (2018) estimated that more than three million girls are at risk of being circumcised annually [2]. The prevalence of the practice differs greatly from region to region [5]. Females who have been affected by it mostly reside in 30 under-developed countries in Africa, the Middle East, and Asia [4]. Furthermore, the WHO (2001) has reported that it has been practiced in the UK, Europe, America, and Australia as a result of immigrants who practice it in their parent countries [6]. The international efforts to end FGM/C practice has been increasing every year since 1997 [7]. Thus, the FGM/C prevalence among (0–14 years old) girls has decreased significantly in many countries [5]. However, the existing trend of the decrease in the practice is not sufficient to overcome the underlying growth in the population, and if the current trend continues in the upcoming years, more young girls will be affected, with an estimate of 68 million girls being subjected to FGM by 2030 [8].

Studies reported that FGM/C practice continues to exist because it is reinforced by customs, culture, beliefs, social pressure, religion, and the assumption that it increases a girl’s chance of marriageability [9,10,11]. This practice is largely performed by male-dominant or patriarchal, patrilineal societies [6]. This is consistent with the studies by Mackie and LeJeune (2009) and Eldin et al. (2018), who described that the practice is motivated by patriarchy and continues to be practiced due to inequality between males and females [12,13]. For example, mothers who oppose it, are unable to abolish the practice because they live in a male-dominated society [14]. However, several studies (El-Dareer (1983); Shell-Duncan and Herniund (2006) and Mackie and LeJeune (2009)) have reported that in many places, it is considered “women’s business” as, in the majority of cases, mothers or grandmothers organize and support the cutting of the daughters [12,15,16]. Primarily female practitioners perform circumcision on other women and girls; however, it is performed to control the bodies of females and their sexuality for the sake of men, as it is perceived to ensure virginity until marriage [17]. The practice is a symbol of social control of female sexual pleasure, and it is linked to the female reproductive role in society [18].

The patriarchal society includes socioeconomic inferiority, social norms, and the desire to get married, as socioeconomic inferiority makes women dependent on marriage for a better situation; this makes them unable to avoid having FGM/C [12]. Also, even if women are economically sufficient, they would still pursue marriage and would be obligated to follow the social norms of patriarchal society. This emphasis on gender norms, social, cultural factors, and power relations drives the persistence of practice [13]. In a patriarchal society, the male might not actively participate in FGM/C decision-making process; however, this does not mean that they have no influence on the practice [19]. When men adopt a passive attitude toward FGM/C, they allow the practice to continue as their silence is seen as approval [15].

Individual or collective decision-making is a process that goes through different stages, occurring over time rather than an instant act. This decision-making process (DMP) is described to be composed of five stages: 1—*knowledge* of the presence of innovative behavior, 2—*persuasion* by forming an opinion, 3—*making a choice* to accept or discard the innovation, 4—*implementing* the innovation, and 5—*confirmation*, when decision-makers try to reinforcement the decision if exposed to conflict [16]. Bjälkander et al. (2012) define the FGM/C’s household decision-making processas to how a family decides whether their daughter should be circumcised or not [20]. Several studies (Shell Duncan et al. (2010); Bjälkander et al. (2012); Kaplan, et al. (2013); and Sabahelzain, et al. (2019)) have cited that the household’s decision of whether to circumcise the girl or not seems to be a result of a complicated process involving multiple individuals and influences [19,20,21,22]. However, there is no published data regarding intervention targeting household decision-makers. 

Previous systematic reviews have not addressed the FGM/C DMP or household decision-makers HHDM(s). The objective of this systematic review is to investigate the FGM/C households’ decision-making process and to identify the main decision-makers in the household. In addition, we assess the need for further research in this field or the possibility of implementing interventions. 

## 2. Material and Methods 

### 2.1. Study Design

This systematic review has included qualitative, quantitative, and mixed methods of primary research studies.

### 2.2. Search Strategy

In line with the Centre for Reviews and Dissemination (CRD) 2009 guidelines [23], an initial review of the existing literature was conducted to justify the need for this systematic review. The initial review of the literature was carried on PubMed, EMBASE, EBSCO CINAHL Plus databases. Then the database of reviews, the Cochrane Database of Systematic Reviews (CDSR), was searched for existing or ongoing systematic review being carried out. Different systematic reviews were found regarding FGM/C; however, no systematic review studied FGM/C DMP or decision-makers. Google Scholar was used to search for studies that are available online but not yet indexed in the database. 

Following the Preferred Reporting Items for Systematic Reviews and Meta-Analyses (PRISMA) guidelines for the systematic reviews, a comprehensive search of the published literature was conducted to identify different publications. The literature search was not limited to a certain country or year and was carried on different databases to avoid missing key studies and to minimize bias. The databases that were searched are PubMed, JSTOR, Ovid MEDLINE, Ovid EMBASE, EBSCO CINAHL Plus, and BioMed Central. SPIDER search strategy tools (Sample, Phenomenon of Interest, Design, Evaluation, Research type) were used to identify the keywords due to its suitability (for a mixed-methods systematic review [24], refer to Table 1).

Booleans operators conjunctions were used to achieve more focused results, and the MeSH browser was used for indexing articles. The keywords that were used to search the databases in combination: “Father” or “mother” or “husband” or “grandmother” or “grandparent” or “household” or “household decision” and (1) “female genital mutilation” or “FGM/C” or “FGM” or “female genital circumcision” or “FGC” or “female circumcision” or “female genital cutting” and (2) “decision-making” or “decision-maker” or “decision-making process” or “attitude”.

Search limits were implemented to refine the scope of the search to primary peer-reviewed articles, available in the English language, and full-text articles.

Furthermore, the scanning of the reference lists of relevant studies that were obtained from the database search was carried out to identify further studies (“reference harvesting”). 

The databases search resulted in 1883 articles, and five articles were obtained from reference harvesting (refer to Figure 1). 

### 2.3. Study Selection

The inclusion and exclusion criteria are listed in Table 2: 

To avoid duplication bias, before the implementation of the inclusion and exclusion criteria, removal of the duplicated articles was done using RefWorks software, and they were manually screened. After the removal of the duplicated articles, the literature search resulted in a total of 1732 papers.

As the topic of this search does not have negative or positive results, the articles are most likely to be representative of all the studies conducted, published, and unpublished. Google Scholar was used to scan for articles in which they were not yet included or conference abstracts. In total, three articles were related to FGM/C decision-making process and decision-makers were found; however, they were excluded because they are not peer-reviewed. Only peer-reviewed articles are included in this search as they present higher quality research and this minimizes bias. 

### 2.4. Inclusion Criteria and Exclusion Criteria Implementation

At the first stage, the articles that resulted from the search after implementing the limits were screened for study design. In the second stage, the title and abstract of the search result were scanned for sample and study designs limited to data collected through interviews, focus groups, questionnaires, and surveys, including qualitative, quantitative, and mixed methods researches. 

From the literature, it was found that in some articles, the decision-making process or decision-makers were not the primary aim and therefore were not mentioned in the abstract, and thus, the “PI” and “E” inclusion and exclusion criteria were implemented at the third stage of screening.

In the fourth stage, 66 relevant studies to FGM/C (S-sample) resulted from the search; their full-text articles were scanned for data collected from participants regarding PI and E criteria, namely, the FGM/C decision-making process and who the decision-maker is regarding whether to cut or not.

Articles that provided insufficient or unfocused information regarding the DMP and/or decision-makers were excluded. After the implementation of the inclusion and exclusion criteria, 23 papers were chosen for the critical appraisal stage (refer to Figure 1 & Table 2).

### 2.5. Data Abstraction

Data were extracted in Microsoft Excel. The data extracted included the in-text citation of the article; the study design; the context of the study, e.g., “area, country”; sample size; if the main aim was related to FGM/C, DMP, and/or HHDMs; the aim of the study; the source of information regarding the DMP and HHDMs namely, household, mother, father, the woman who underwent FGM/C, healthcare professionals, or others; the results related to FGM/C DMP and/or HHDMs; the limitations of the study. 

### 2.6. Analysis

This systematic review contains data from both qualitative and quantitative research, and therefore, meta-analysis could not be conducted. Microsoft Excel was used for organizing and analyzing the data extracted from the included articles. Then, a textual narrative synthesis was carried out.

#### 2.6.1. Critical Appraisal

The critical appraisal was applied to the 23 studies to examine the studies’ methodological strengths, weaknesses, the validity of the research, the trustworthiness of the results, and the presence of biases (refer to Table 3). It was also done to test whether the studies have been designed, carried on, and written in a reliable matter and whether they provide a meaningful answer to this systematic review question. The studies were appraised using different types of appraisal tools, where the qualitative research’s quality assessment was conducted using the Critical Appraisal Skills Programme (CASP) [25]. Cross-sectional studies were appraised using the Appraisal tool for Cross-Sectional Studies (AXIS), which was developed specifically for the appraisal of this type of design [26] (refer to Table 4). This review included ethical appraisal, which carried out to improve the ethical and methodological quality of this review by avoiding papers with ethical inadequacies. 

#### 2.6.2. Ethical Statement

Application for ethical approval was made to the School Research Ethics Panel (SREP) Allied Health, Anglia Ruskin University, which concluded that no ethical approval is needed for this research as this systematic review only require to retrieve and synthesize of data from already published articles.

#### 2.6.3. Outcome of the Critical Appraisal and Ethical Appraisal

The critical and ethical appraisal resulted in 17 studies that were included in the review. Six studies were excluded from this systematic review for the following reasons: low internal validity of the study, which affects the reliability of the results (one study [27]); low validity and reliability of the results, in addition to the lack of ethical considerations (two studies [14,28]); ethical appraisal due to lack information regarding participant consent, confidentiality, and obtaining ethical approval (three studies [29,30,31]). 

## 3. Results

### 3.1. Characteristics of the Included Studies

The review included 17 studies from 11 different countries; 13 of the studies were conducted in the African countries of Ethiopia, Sudan, Somaliland, Sierra Leone, Guinea, Gambia, and Senegal; two studies were conducted in Mideastern country Kurdistan regions in Iraqi; two studies were carried among immigrant communities, one in the United States of America, and the other in Canada. Only four of the studies’ main aim was to explore the DMP and/or HHDMs [19,20,22,32]. The remaining studies’ [33,34,35,36,37,38,39,40,41,42,43,44,45] main objectives did not include the DMP and/or HHDMs; however, they collected sufficient data regarding the DMP and/or HHDMs from their samples.

### 3.2. The Designs of the Included Studies

Nine qualitative studies were included, in which data were collected using either in-depth interviews or FGDs or both. Furthermore, eight cross-sectional studies—community-based, school-based, household surveys—were included where data were collected using a self-administered questionnaire, face-to-face interview, or audio computer-assisted self-interviews. Two of the cross-sectional studies collected both qualitative and quantitative data. 

### 3.3. Source of Information Regarding the DMP and/or HHDMs

The data extracted from the studies were self-reported by the participants in the studies mentioned. The studies included various samples differing in age groups, social positions, and education levels. Some samples reported on their own experience regarding FGM/C, while others reflected on their community’s DMP and decision-makers. 

Characteristics, designs, sources of information, and summaries of the findings are presented in Table 5.

The most cited FGM/C HHDMs by the samples of the reviewed articles are mothers, fathers, and grandmothers. Three main themes regarding the DMP and HHDMs emerged during the analysis: (1) the decision-making process, (2) are females the main decision-makers?, and (3) the role of fathers as decision-makers. The available data shed light on the decision-making process within the household and the community, revealing the complexity of the process and the shift that took place over time. The themes below reflect the variation of the DMP among different regions that practice it. The data shed light on the main FGM/C household decision-makers and the power-relations in regard to the DMP.

### 3.4. Decision-Making Process

Some studies examined the time-shift in the FGM/C DMP, such as studies in Senegal and Gambia [37] that examined how overtime large group circumcisions were replaced with private circumcisions within the family, which shifted the decision-making from “when” to circumcise to “whether” to circumcise, taking the power of decision-making from the community and handing it to the family. Additionally, the paper reveals that inter-ethnic marriages between one family practicing FGM/C to another who does not lead to a debate regarding the decision of circumcising the daughter. Another study revealed the differences between young parents’ generation and the generation of grandparents and reported more involvement of young men in the DMP than the grandparents’ generation [41]. 

Several studies established that when it comes to decision-making, the process involves several members of the nuclear family, extended family, and acquaintances. There is involvement of the mothers, co-wives, grandmothers, aunts, and fathers [32], and when there is conflict, each individual has a different degree of power over the decision. The young females usually have limited authority in the decision compared to older women, but they can strengthen their opinion by asking the support of senior females in the family who have more power. The study participants from Sudan shared similar findings regarding younger women having less power than older women, who insist on FGM/C [34]. In another study of Keita and Blankhart (2001) in Guinea [33], among Malinke families, the FGM/C decision is taken by agreement among all the adult members of the family, extended family, and members of the social circle. Sabahelzain et al.’s (2019) household surveys conducted in Sudan indicated that the FGM/C HHDMP involved discussions among the nuclear and extended family and non-family members, including “mothers, fathers, maternal grandmothers, paternal grandmothers, aunts, professional or anti-FGM/C activists, sons, daughters, and uncles”, with different degree of participation among them [22]. In Sierra Leone, participants shared that the decision involves “mothers, grandmothers, aunts, and fathers” and, to a lesser extent, grandfathers, husbands, guardians, sisters; two of the participants even reported themselves as the ones who made the decision to undergo FGM/C [20].

This review reveals the comparison of the DMP in the Iraqi Kurdistan region with the DMP in African countries, where the decision to circumcise the daughter is only made by closed females family members and that males in the family are not included in the decision nor informed about the circumcision [38,40]. 

### 3.5. Are Females the Main Decision-Makers?

According to a study in the Somali and Harari society of eastern Ethiopia, mothers are the decision-makers for female circumcision and play a major role in the practice. The mothers desire to circumcise their daughters to optimize their future prospects due to their own fear of violating the tradition [37]. A more recent study in the same area [43] confirmed their previous findings, where the majority of the sample cited their mothers as the decision-makers for their circumcision, followed by their grandmother, while a small percentage claimed that their fathers were the decision-makers. The study also reported that in 5% of the sample, the girls themselves made the decision to undergo FGM/C. Similarly, another study among Canadian-Somali participants who reported their own FGM/C experiences said that their mothers were responsible for arranging their FGM/C, with the exception of one participant who cited her grandmother [39]. The study reported silence regarding the fathers’ role in decision-making, and that the fathers and uncles were away or disagreed with their partner regarding the FGM/C of the daughter. The sample from Somalian midwives reported mothers as the first FGM/C decision-makers, and that, generally, the mother and grandmother in a family proposed the idea, while the father pays the practitioner [35]. However, in case of disagreement in the decision, the final decision is carried out by the father. Another study in Somaliland shared [36] that most of the participants claimed the mother is the main decision-maker regarding FGM/C with a small percentage of participation in the decision-making by fathers. The study also supports the finding that the mothers’ decision is influenced by the social structure of the community and emerges from her desire to ensure better chances for marriageability for her daughter and fear of being shamed by society. A study of Erbil, Iraq Kurdistan, likewise indicated that mothers or grandmothers are the main decision-makers, while fathers are not involved in the decision [40]. These findings are supported by another research conducted in the same location, which also reported that the circumcision of the girl is done without the knowledge of fathers or brothers as it is a matter of females, i.e., “mothers, grandmothers, aunts” [38]. In the study in Guinea, Keita and Blankhart (2001) stated that religious leaders and community samples viewed FGM/C as a women affair, with a small majority of interviewees stating that the women in the family are the ones responsible for the decision to carry out FGM/C, a quarter of the sample citing men, and a fifth of the sample citing “others” as the decision-makers [33].

### 3.6. The Fathers’ Role as Decision-Makers

Although several articles mentioned that FGM/C as women’s business, other articles have mentioned fathers as the sole decision-makers or part of the DMP.A study in Sierra Leone found fathers as the solo FGM/C decision-maker were as equally mentioned as mothers, while some participants reported both parents taking the decision together [20]. It was also reported that in almost two-thirds of cases, fathers were involved in the decision-making process when the final decision was to leave the girl uncut, while their involvement was less than a third when the final decision is to cut the girl [22]. Almroth et al. (2001) reported that fathers were more involved in the decision-making when the decision was not to perform FGM/C and, in some cases, the final decision not to circumcise was made only by the father [41], while the young male sample in Berggren et al.’s (2006) study described that they took the FGM/C final decision, opposing their wives not to circumcise their daughters [34].

A study found that less than the majority of men participate in the DMP, especially if they were single, and less than a tenth of fathers act as the final decision-makers [19]. However, the study recognized that men who come from families that do not practice FGM/C do not intend to circumcise their daughters, and expect the same attitude from their family. In contrast, for men who come from families that practice FGM/C, the majority of them appear to be supportive of the practice, intend to circumcise their daughters, and they expect their families to share their attitude. Therefore, they do not participate in the decision-making process on the assumption that the final decision is similar to their own unspoken decision; additionally, females are unable to make the decision not to circumcise their daughter without the active support of their husbands. On the other hand, research in Bale Zone, Ethiopia, discovered that most of the participants identified both parents as the decision-makers; however, they rarely identified the father as the only decision-maker (1.6%). Although it might be the mother who introduces the practice, it is the father that facilitates the action of the practice beforehand [43]. Another study in Addis Ababa, Ethiopia, found that almost a quarter of the sample cited only fathers as decision-makers, while one-third of the sample claimed only mothers and the rest of the decisions were made by either parents or relatives [42]. Meanwhile, among West African immigrants in the USA, mothers followed by maternal and paternal grandmothers were the most cited decision-makers by both male and female participants, while forty percent of the sample cited fathers as decision-makers [44]. Finally, although the research by Isman et al. (2013) cited females as the main decision-makers, they also reported that when there is a different opinion regarding performing FGM/C, the father of the girl appears to be the final decision-maker [35].

## 4. Discussion

The FGM/C household decision-making process is complex that differs from one area to another and changes with time. Putting into consideration that FGM/C is an old tradition in which some communities (such as African) that practice it adhered to it a long time ago, those communities have already passed by all the phases of the decision-making process and established their decision and implementation process [16]. Therefore, when those communities receive new information regarding the practice, some of them re-evaluate their decision bypassing the five stages of the decision-making process [16] all over again (1—gain new knowledge about the FGM/C practice, 2—forming either an inclination to change the practice or continue with the tradition, 3—making a choice depending on the inclination, 4—implementing their choice, 5—when faced with debate, seek reinforcement of the decision). Some members might decide at any phase not to change their practice and follow tradition, as dictated by the social norms.

In some communities, e.g., Sierra Leone, Sudan, Gambia, Senegal, Guinea and Egypt, where there is open dialog regarding this practice, especially when part of the community have abandoned the practice, the DMP first three stages seems to involve several individuals from close family and others [20,22,31,32,33]. On the other hand, the sensitivity of FGM/C topic among other communities, e.g., Iraq Kurdistan, Somalia, Somali, and Harari regions, affect the possibility of opening dialog regarding the practice between the sexes; therefore, making the FGM/C decision a women’s responsibility [36,38,39,40,43], where the mothers have full responsibility of their daughters and fathers are responsible for the males [37]. However, the statement that females, especially mothers, are the ones controlling the DMP in these communities should be treated with caution. Although fathers are not the main decision-makers, they appear to facilitate the practice [43] by maintaining passive attitudes when they have the power to influence the decision [36] or paying for the circumcision. Moreover, fathers have the power to make the final decision, and they appear to exercise their power when they oppose circumcising their daughters [34,35].

Females’ or mothers’ decisions are influenced by their desire to maintain tradition and fulfill their obligations within the society by ensuring the marriageability of their daughters where circumcised females are more favorable for marriage [36,37,43]. Furthermore, mothers might hold low power in decision-making and limited authority to contest to the practice in opposing to older females who support tradition and hold higher power in the community and within the family [32]. The mothers’ desire to maintain tradition could be caused by the overall patriarchal context of the society and the unequal power relations rooted in the maternalistic relations between females and their mothers [34]. However, researchers [15] argued that the decision-making power is not fixed and can change over time, which can explain the findings of different studies [22,41,46] reported women with high education and possibility to work outside tend to decide not to circumcise their daughter, as education and financial stability increases the woman’s decision-making power within the household.

The results of two studies (Amusan and Asekun-Olarinmoye 2008; Garba et al. 2012) conducted in Nigeria mentioned contradictory findings and claimed that within the majority, males in the family are the FGM/C decision-makers, who often request the circumcision to be performed, with the paternal grandfather being the main decision-maker, followed by the father and maternal grandfather [14,30]. This contradiction can be explained by the variation of the DMP and HHDMs between countries and regions due to the differences in the social structure.

While examining the role of men in the decision-making process and as decision-makers, the findings of this systematic review support that fathers tend to have a more active role in participating in the decision-making and act as decision-makers when the decision is not to circumcise the daughter [22,41]. The men do not participate in the decision-making process when their attitude toward the practice is similar to that of their families [19], and this emphasizes the fact that females in a patriarchal society are unable to leave their daughters uncircumcised without the active support of the father. Thus, men might not be actively participating in FGM/C decision-making process, but they are still decision-makers. Some studies [41,42,44] mentioned higher participation among males in the DMP, which can be due to high exposure to anti-FGM/C campaign, their sample being composed of educated males, or immigrants that live outside their country of origin.

Healthcare professionals working in gynecology and obstetrics can encourage open conversation regarding FGM/C decision-making by play an essential role in engaging partners who come to healthcare facilities in active dialogues about the practice, especially if the female partner underwent the practice herself. Thus, it is essential to provide education of all aspects of FGM/C, including the DMP to gynecologists, obstetrics, midwives, and healthcare workers, in both developed and undeveloped countries. FGM/C has severe clinical implications on women because it affects their sexual, psychological, and reproductive health; furthermore, it puts the lives of the mother and child at risk during delivery.

### Strengths and Limitations

To the best of the authors’ knowledge, this study is the first systematic review conducted regarding the FGM/C decision-making process and decision-makers that identifies the DMP and key decision-makers. To minimize bias, the critical appraisal was carried out on two different occasions with a one-week period between the first and second appraisals, and then the two appraisals for each study were compared. The study only included peer-reviewed articles, which is considered a strength as well as a weakness. The review only includes articles published in the English language, which introduce location and language bias. Additionally, excluding articles that are not available in full-text can cause articles that are up-to-date to be missed out. The quality assessment and data extraction were carried out by the first author only; cross-checking was not conducted, which might have affected the quality of reporting and the analysis. Finally, variation in the collected data can be observed as it is caused by the different range of samples in the studies reviewed.

## 5. Conclusions

This review is the first to provides summarized information about the DMP, HHDMs, and FGM/C. The most cited FGM/C HHDMs by the samples of the reviewed articles are mothers, fathers, and grandmothers. This review introduces the presence of silence when discussing FGM/C among the sexes, caused by social and cultural obligations that influence the DMP. This causes confusion in regards to who has more authority in the DMP.

It was shown that the DMP is an important issue in FGM/C in different countries and ethnicities. However, due to limited studies that focus on the topic, further research is needed. 

The findings of this review show that active participation of fathers in the decision-making process could free mothers from the social-pressure and responsibility of carrying on traditions, creating a more favorable environment for FGM/C abandonment. Designing and implementing community programs that involve males and females in open conversations regarding the practice, and education programs for men regarding their important role in DMP may also assist in ending the practice.

## Figures and Tables

**Figure 1 ijerph-17-03362-f001:**
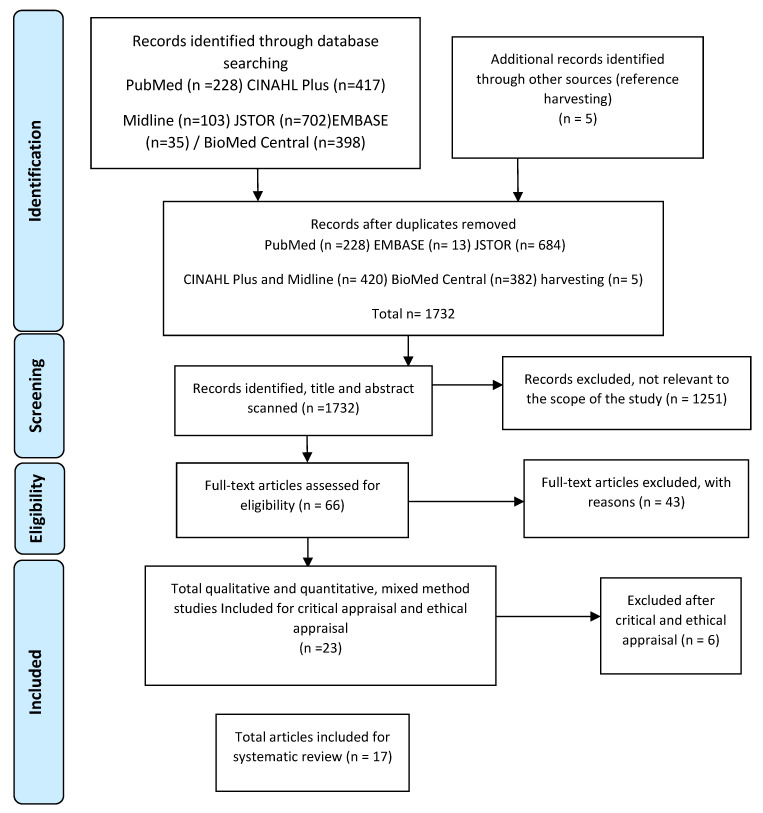
Preferred Reporting Items for Systematic Reviews and Meta-Analyses (PRISMA) 2009 flow diagram.

**Table 1 ijerph-17-03362-t001:** SPIDER search tool.

S—Sample	Households or individuals from a community that practice FGM/C
PI—Phenomenon of Interest	FGM/C decision-making process
D—Design	Interview, Focus Group Discussions (FGDs), questionnaire, survey
E—Evaluation (Outcome)	Decision-makers
R—Research type	Qualitative, quantitative, and mixed methods

**Table 2 ijerph-17-03362-t002:** Inclusion and exclusion criteria.

	Inclusion	Exclusion
**Sample (S)**	Households and individuals who are involved in FGM/C practice All countries	Any other topic regarding FGM/C example; intervention and policies, FGM/C reconstructive surgeries.
**Phenomenon of Interest (PI)**	FGM/C decision-making process	Did not collect data regarding FGM/C decision-making process
**Design (D)**	Interview, focus groups, questionnaire, survey	Intervention research, Case study
**Evaluation (E) “Outcome”**	FGM/C decision-makers	Any article that did not collect data regarding the FGM/C decision-makers in the household
**Research type (R)**	Peer-reviewed Primary research Qualitative studies Quantitative studies Mixed methods studies No timeframe Published in the English language	Not peer-reviewed Languages other than English Short Commentaries/Viewpoints Literature other than primary research

**Table 3 ijerph-17-03362-t003:** Critical appraisal for qualitative studies using the Critical Appraisal Skills Programme (CASP) tool.

Qualitative Studies: CASP Tool	Section A: Are the Results Valid?					Section B: What Are the Results?				
Reference	Was There a Clear Statement of the Aims of the Research?	Is a Qualitative Methodology Appropriate?	Was the Research Design Appropriate to Address the Aims of the Research?	Was the Recruitment Strategy Appropriate to the Aims of the Research?	Was the Data Collected in a Way that Addressed the Research Issue?	Has the Relationship between Researcher and Participants Been Adequately Considered?	Have Ethical Issues been Taken into Consideration?	Was the Data Analysis Sufficiently Rigorous?	Is There a Clear Statement of Findings?	How Valuable Is the Research?
Vissandjée et al. 2003 [28]	+/-	+	-	+/-	-	-	-	-	+/-	-
Shell-Duncan et al. 2018 [32]	+	+	+	+/-	+/-	+/-	+	+	+/-	+
Keita and Blankhart, 2001 [33]	+	+	+	+/-	+	-	+/-	+/-	+/-	+/-
Berggren et al. 2006 [34]	+	+	+	+	+	+/-	+	+	+/-	+/-
Isman et al. 2013 [35]	+	+	+	+/-	+	+	+	+	+	+
Lunde and Sagbakken 2014 [36]	+	+	-	+/-	+/-	+/-	+	+	+/-	+/-
Abathun, Sundby and Gele, 2016 [37]	+	+	+	+	+	-	+	+/-	+	+
Shabila, Ahmed and Safari 2017 [38]	+	+	-	+	+/-	+	+	+	+	+
Jacobson et al. 2018 [39]	+	+	+	+/-	+	+/-	+	+/-	+	+/-
Ahmed, Shabu and Shabila 2019 [40]	+	+	-	+	+	-	+	+/-	+	+

(+) = item adequately addressed, (-) = item not adequately addressed, (+/-) = item partially addressed.

**Table 4 ijerph-17-03362-t004:** Critical appraisal for cross-sectional studies using the Appraisal tool for Cross-Sectional Studies (AXIS).

Reference	Introduction	Methods																
	Were the Aims/ Objectives of the Study Clear?	Was the Study Design Appropriate for the Stated Aim(s)?		Was the Sample Size Justified?		Was the Target/ Reference Population Clearly Defined? (Is It Clear Who the Research Was about?)	Was the Sample Frame Taken from an Appropriate Population Base So That It Closely Represented the Target/ Reference Population under Investigation?		Was the Selection Process Likely to Select Subjects/ Participants That Were Representative of the Target/ Reference Population under Investigation?		Were Measures Undertaken to Address and Categorize -non-Responders?		Were the Risk Factor and Outcome Variables Measured Appropriate to the Aims of the Study?		Were the Risk Factor and Outcome Variables Measured Correctly Using Instruments/ Measurements That Had Been Trialled, Piloted or Published Previously?		Is It Clear What was Used to Determine Statistical Significance and/or Precision Estimates? (e.g.,*p*-Values, Confidence Intervals)	Were the Methods (Including Statistical Methods) Sufficiently Described to Enable Them to Be Repeated?
Garba et al. 2012 [14]	+	+		-		+	-		-		NA		-		-		+	+/-
Kaplan et al. 2013 [19]	+	+		-		+/-	+/-		-		-		+		+		+	+
Bjälkander et al. 2012 [20]	+	+		-		+	+/-		+		-		+		+		-	+
Sabahelzain et al. 2019 [22]	+	+		+		+	+		+		+		+		+		+	+
Gebremariam, Assefa and Weldegebreal 2016 [27]	+	+		+		+/-	+/-		+/-		+		-		+/-		+	-
Herieka and Dhar 2003 [29]	+	+		-		+	+		+		-		+		-		+	+
Amusan and Asekun-Olarinmoye 2008 [30]	+	+		+		+	+		+		-		+		+		-	+
Tag-Eldin et al. 2008 [31]	+	+		+		+	+		+		NS		+		+		+	+
Almroth et al. 2001 [41]	+	+		-		+	+		+		+/-		-		-		+	+/-
Shay, Haidar and Kogi-Makau 2010 [42]	+	+		+		+	+		+		-		-		+		+	+/-
Bogale, Markos, and Kaso 2014 [43]	+	+		+		+	+		+		+		+		+		+	+
Akinsulure-Smith and Chu 2017 [44]	+	+		-		+	+		+/-		-		+		+/-		-	+
Abathun, Sundby and Gele 2018 [45]	+	+		+		+	+		+/-		+		+		+		+	+/-
Reference	Results											Discussion				Others		
	Were the Basic Data Adequately Described?		Does the Response Rate Raise Concerns about -non Response Bias?		If Appropriate, Was Information about -non Responders Described?			Were the Results Internally Consistent?		Were the Results Presented for All the Analyses Described in the Methods?		Were the Authors’ Discussions and Conclusions Justified by the Results?		Were the Limitations of the Study Discussed?		Were There Any Funding Sources or Conflicts of Interest that May Affect the Authors’ Interpretation of the Results?		Was Ethical Approval or Consent of Participants Attained?
Garba et al. 2012 [14]	+		NA		NA			+		+		+		-		-		+/-
Kaplan et al. 2013 [19]	+		+		NA			+		+		+		+/-		+/-		+
Bjälkander et al. 2012 [20]	+		+		+			+/-		+		+		+		+		+
Sabahelzain et al. 2019 [22]	+		+		+			+		+		+		+		+		+
Gebremariam, Assefa and Weldegebreal 2016 [27]	+		+		-			+		+/-		+		-		+		+
Herieka and Dhar 2003 [29]	+		+		-			+		+		+		-		NS		NS
Amusan and Asekun-Olarinmoye 2008 [30]	+		+		NA			+		+		+/-		-		NS		-
Tag-Eldin et al. 2008 [31]	+/-		NS		NS			+		+		+		-		-		-
Almroth et al. 2001 [41]	+		+		+			+/-		+		+		-		-		-/+
Shay, Haidar and Kogi-Makau 2010 [42]	+		+		NS			+		+/-		+/-		+		+		+
Bogale, Markos, and Kaso 2014 [43]			+		-			+		+		+		+		+		+
Akinsulure-Smith and Chu 2017 [44]	+		NA		NA			+		+		+		+		NS		+
Abathun, Sundby and Gele 2018 [45]	+		+		NS			+		+		+		+		+		+

(+) = item adequately addressed, (-) = item not adequately addressed, (+/-) = item partially addressed, NS= not stated or “I do not know”, NA= not applicable.

**Table 5 ijerph-17-03362-t005:** Data extraction table (characteristics of the 17 papers included in the review and summary of their findings).

Reference	Study Design and Methods	Context	Sample Size	Is the Aim Specific to the DMP/HHDMs?	Aim of the Study	Source of Information Regarding FGM/C DMP and HHDMs	Key Findings Regarding the DMP and/or HHDMs	Limitation
Abathun, Sundby and Gele 2016 [37]	Qualitative study using FGDs	Somali and Harari region, Ethiopia	64 women and men participants, 8 participants perFGDs	No	Explore the attitude toward the practice of FGM/C	Participants reflecting on their community.	In Somali communities’ mothers are responsible for daughters and fathers for sons. Mothers play essential role in FGM/C in both Harari and Somali regions. Mothers are the HHDMs to FGM/C in both the regions.	Inability to generalize the findings of this study to the population. The effect of the interpreter on the data.
Abathun, Sundby and Gele 2018 [43]	Cross-sectional quantitative study using interviews	Jigjiga town, Somali region and Harar town, Harari Region, Ethiopia	480 Girls and Boys (16 to 22 years old)	No	1—Investigate pupil’s perspectives toward the abandonment of FGM/C. 2—investigate the source of information that impact pupils’ attitude toward the practice.	All participants reflected on main decision-maker regarding FGM/C in their region. Female students who underwent FGM/C (N=79) reflected on who made the decision to circumcise them.	41.2% of all the participants cited mothers as the decision maker to perform FGC in the regions, followed by both parents (34.5%), and fathers (5.2%). Among girls who underwent FGM/C, 67.1% stated that the decision was made by the mother (67.1%), followed by their grandmothers (19%), then father (8.9%), 5.1% reported that they made the decision.	The self-reported answers; can cause social desirability bias. The study was conducted among students; leading to inability to generalize the findings due to high illiteracy rate in the study area.
Ahmed, Shabu and Shabila 2019 [40]	Qualitative study using FGDs	Erbil governorate, Iraqi Kurdistan Region	Six FGDs including 51 women (age 18 and above)	No	1—Assessing the knowledge, beliefs, and attitude of a sample of Kurdish women of FGM/C. 2—Identifying the main enabling factors for performing FGM/C and the barriers to ending it.	Circumcised women	The participants stated that the mother or grandmother usually decides to circumcise the girls, without the involvement of fathers or men in DMP.	Underestimation of statements. This study is limited to participants from Erbil governorate. Response bias; participants might not have expressed their true attitude toward FGM/C.
Akinsulure-Smith and Chu 2017 [44]	Cross-sectional /survey using audio computer-assisted self-interviews	New York City, NYC, United States of America	107 West African immigrants (36 male and 71 female) over the age of 18.	No	Exploring the knowledge and attitudes toward FGM/C by African male immigrants living in NYC.	Male and female participants reporting who they felt were the primary HHDM	Both male and female participants most commonly reported mothers as HHDMs (65.6%). Maternal and paternal grandmothers were the next most commonly cited HHDM (66.7% of women and 40% of men cited maternal grandmothers, 56.7% of women and 33.3% of men cited paternal grandmothers). Fathers were cited as HHDM by 40% of all participants.	Limited generalization of findings. Self-selection bias. It focused on subgroups based on country of origin, rather than on ethnicity. The attitudes may have reflected a long period away from country of origin, social norms, and influence of who and what dictate the practice.
Almrothet al. 2001 [41]	Cross-sectional- Community based survey using interviews	Village in Gezira scheme, Sudan	120 young parents and grandparents.	No	1—Investigating the practice of FGM/C in a rural area in Sudan. 2—Determining the factors influence this practice among men and women of a young parental generation and a generation of grandparents.	Young parents (married women 30 years old and below, married men 35 years old and below, or the eldest child is 4 years old or below or no children) and grandparents (independent of age)	54 out of 120 said mothers were the HHDMs of FGM/C. The girl’s father was more involved when the final decision was not to perform FGM/C. Young fathers were more involved in the DMP in comparison to past generations, especially when the decision was not to perform FGM/C	*The findings are specific to Gezira scheme and not generalizable to other areas.
Berggren et al. 2006 [34]	Qualitative study using interviews	Khartoum State, Sudan	22 in-depth interviews (12 women and 10 men)	No	Exploring Sudanese women's and men's perceptions and experiences of FGM/C with emphasis on reinfibulation.	Participants (women aged 19–68 years old, and 10 men aged 28–47 years old)—younger or older age groups were not explained	Younger females stated that older women are the ones with power and the ones insisting on FGM/C, preferably type III. The few older women admitted their interference and stressed that it is an important tradition. Several of the men claimed that they were hardly involved at all in the decision to perform FGM/C. However, a few men described how they opposed their wives and taken the decision themselves, not to circumcise their daughters.	Men participants were more educated than average. Also, as the interviewers were females; when women interview men, there might be a tendency to present a favorable image, the socially desirable response that refers to giving the answers that are consistent with prevailing social mores.
Bjälkander et al. 2012 [20]	Cross-sectional community-based survey using interviews	Northern province of Sierra Leone	350 girls (10–20 years old)	Yes	Identifying decision-makers for FGM/C and the extent of medicalization of the practice in Sierra Leone.	The girls who were cut (N=190) reflecting on their own experience, if they did not know, the parent or guardian reported for them.	Females reportedly dominate the decision-making process; however, fathers (n=54) were mentioned equally often as mothers (n=51) as the HHDMs. Other decision-makers mentioned in responses were grandmothers (n=39) and aunts (n=29) and, to a lesser extent, grandfathers, husbands, guardians, grandmothers and mothers jointly, and sisters. In seven instances, a combination of relatives made the decision for FGM/C: in six cases, it was the mother and father together, and in one case, it was the mother and grandmother together. Two girls reported themselves as the decision-makers.	The sample participant is not representative of the population in Sierra Leone, as it does not cover all ethnic groups and has geographical limitation. The methodology of the study did not aim to address the whole decision-making process.
Bogale, Markos, and Kaso 2014 [43]	Community-based cross-sectional study, quantitative methods enhanced with qualitative methods, using questionnaires, in-depth interviews and FGDs.	Bale zone, Southeast of Ethiopia	634 child-bearing age women; four FGDs (8 participants per FGD) and 8 interviews	No	Assessing the current prevalence of FMG, its health consequences and factors underpinning the perpetuation of this practice	Respondents	57.5% of respondents cited both mothers and fathers as FGM/C HHDMs for their daughter; 37.3% identified HHDMs as only mothers; 1.6% only the father, and 3.5%others, i.e., “the girl herself, grandparents, other relatives, and neighbors’’. In-depth interview findings reported that all the family members are responsible of a girl circumcision. ‘Even if only a mother presents on an operation, a father also facilitates things beforehand’.	Response bias; FGM/C is a sensitive and stigmatizing social issue in the study area; in addition to the likelihood for women to give culturally acceptable answers to the interviewer, which can lead to respondents’ bias.
Isman et al. 2013 [35]	Qualitative research using interviews	Somaliland	8 midwives	No	Elucidate midwives’ experiences in providing care and counseling to women problems related to FGM/C.	Midwives reflecting on their community	All midwives agreed it is first and foremost the mother in the family who decide regarding FGM/C. The mother and grandmother propose the idea, and the father pays the practitioner. The fathers could have a say regarding if and what type of FGM/C to be performed, but mothers exert a strong influence on the decision. However, if there are different opinions, the father is who takes the final decision. Other members of the family have minor impact on influencing the decision.	Information bias; all interviewed midwives were trained in working with care and counseling of women affected by FGM/C. While being asked about their personal opinion concerning FGM/C, they might have felt uncomfortable to reveal positive feelings or attitude towards it. The small study sample population could be seen as a limitation as the findings are not generalizable.
Jacobson et al. 2018 [39]	Qualitative study using interviews	Toronto, Canada	14 Somali Canadian women (21–46 years old), who underwent FGM/C	No	1—Understand Somali-Canadian women’s experiences of FGM/C. 2—How do they experience their bodies in their current, Canadian lives.	Women reflecting on their own experience	participants reported that their mothers were who arranged the FGM/C procedure and determined when was the right time to perform it, while one participant said that her grandmother was who decided when to have FGM/C. Silence was observed with regard to the father’s role in decision making, among other topics. Participants reported that their fathers and uncles were away or disagreed with the FGM/C decision.	Results may not be generalizable to newly-arrived Somali immigrant populations, or other populations with FGM/C in a general western context. Despite the presence of interpreters, language nuances always exist.
Kaplan et al. 2013 [19]	Cross-sectional survey, quantitatively done using interviews	Lower River Region, North Bank Region, and West Coast Region of Gambia	993 men (16 years old and above)	Yes	Exploring the knowledge and attitudes of Gambian men towards FGM/C, as well as practices in their family and household.	The participant reported on their own experience. 694 men responded to DMP question. While 662 men participants answered the question of the final decision-maker to practice FGM/C on their daughter	34.8% of men take part in this DMP, their participation is less if they are single (married 39.3%, single 21.1%). Only 8.0% of men take the final decision towards subjecting their daughters to the practice, and 6.2% join the wives in this decision. 75.8% cited women as FGM/C decision-makers, and 10% reported that itis a decision of other relatives and community members.	The sensitivity of the FGM/C topic leading to resistance to talking openly. Serahule’s ethnicity sample size was small in comparison to other ethnicity samples.
Keita and Blankhart 2001 [33]	Qualitative community-based study using interviews and FGDs	Faranah District, Guinea	482 men and women were interviewed, and 22 FGDs with mostly older women and community leaders	No	To identify current main factors motivating FGM/C practice and other factors that might help to bring change.	Women from different age group, married men, community and religious leaders, traditional practitioners and health workers	Small majority of interviewees counted women in the family as those who were in charge of the decision to carry out FGM/C, as it is considered women's affair. This view was mainly among religious and community leaders; 25% of interviewees cited men as those who decide regarding FGM/C within the family, and nearly 20% said that the decision-maker was either the aunt, the mother, the husband, the two parents together, or someone else from the same social group.	* Convenience sampling, selection bias. Relationship between interviewers and interviewees was not identified, possibility of response bias or information bias. The findings are not generalizable to other populations.
Lunde and Sagbakken 2014 [36]	Qualitative study using in-depth interviews and observation	Hargeisa, Somaliland	22 organizations representatives, 5 nurses/midwives, 2 traditional birth attendants, 9 lay society representatives	No	1—Assessing current conceptions of FGM/C and efforts to stop the practice. 2—Assessing opinions on FGM/C abandonment	The sample	Most of the sample participants claimed that it was the mother’s role to decide if, when, and how FGM/C should happen. However, younger females stated that even though mothers have the primary responsibility for FGM/C, fathers can influence the decision. Extended family and social structures can have influencing roles in the DMP.	Limited generalization of the findings to the population.
Sabahelzainet al. 2019 [22]	Community-based cross-sectional household survey—mixed methods study using interviews	Khartoum and Gedaref States, Sudan	515 households	Yes	Investigating the FGM/C DMP and the role of gender power relations in Sudan	One family member was interviewed from each household and reported on the household’s decision	HHDM on FGM/C involved discussions among several members. In around 75%of the DMP, mothers were involved. A greater proportion of fathers was involved in household discussions where the final decision was to leave the daughter uncut (65%) than to cut (28%). A greater proportion of maternal grandmothers (31%) were involved in households DMP that decided to cut the youngest daughter than in households that decided to leave her uncut (5%), paternal grandmothers were involved with (16.8%) when the decision is to cut, while aunts involvements were around 7.4 to 7.5%. About one in five households (21%) that decided to leave their daughter uncut reported that a profession or activist was involved in DMP. Others that were involved included sons, daughters, and uncles.	Conducted in only two states, which limits the generalization of the study. Causal inferences cannot be made. Majority of participants were female, which may not reflect the views of men and introduce unintentional bias. Other household members were present during some of the interviews, which may have introduced response bias.
Shabila, Ahmed and Safari 2017 [38]	Qualitative study using in-depth interviews	Erbil city, Iraqi Kurdistan Region	21 obstetrician/ gynecologists, nurses, and midwives (all women)	No	Assessing the FGM/C knowledge, attitude, and personal and professional experience of health professionals.	Participants reflecting on their experience and their society	The sample participants agreed that it is generally grandmothers or mothers who make the decision to circumcise the daughter. The male family members and father are usually not informed or involved in DMP	*findings cannot be generalized to the population as the study focus on health professionals.
Shay, Haidar and Kogi-Makau 2010 [42]	Cross-sectional using self-administered questionnaire	Addis Ababa, Ethiopia	442 study subjects; “the questionnaire was answered by the parents or families of the study subjects.”	No	1—Prevalence of FGM/C among primary school girls. 2—Assessing the driving factors behind FGM/C among.	Parents or families of girls with FGM/C (N=106; among 442 samples only 106 girls underwent FGM/C)	The decision to subject the girl to FGM/C was most frequently made by mothers (38.7%), compared to fathers (24.5%), both parents (22.6%) and relatives (14.2%).	The urban population is not representative of the rural population. The findings were based on self-reporting, which might have biased the information.
Shell-Duncan et al. 2018 [32]	Qualitative study using FGDs	Senegal and The Gambia	15 FGDs with6±8 women (age 18 and older) in each group	Yes	Explore the social norms and dynamics that influence decision making regarding FGM/C in Senegal and The Gambia.	Younger (under 30 years of age) and older women (over 30 years of age) reporting on the shift in DMP.	Large group circumcisions have become less common, and the decision making regarding when and how to circumcise has shifted to the family. The increase in inter-ethnic marriages between ethnicity that practice FGM/C to others who do not practice it, complicated decision-making, and cause debates about modifying or ending FGM/C. Several members of the extended family, mothers, co-wives, grandmothers, aunts, and fathers, participate in decision making. In case of conflict, these individuals have different degrees of power regarding the decision.	The research sites do not provide nationally-representative sample.

*Limitation of the study was not stated in the article.

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
