# Peer review of "Decision-Making Process in Female Genital Mutilation: A Systematic Review"

_ijerph, 2020, doi:10.3390/ijerph17103362_

Round 1
Reviewer 1 Report
Good focus on an understudied public health issue. However entire article needs a proof reading my a scientific writing expert. There are a lot of grammatical errors and writing issues in terms of conciseness, clarity and flow. Please consult a writing expert and get the entire manuscript corrected.
Abstract – FGM/C abbreviation needs to be explained at first place. What is /C? Poorly worded abstract. Please rectify all grammatical errors.
Background:
The background needs a lot of work in terms of clarity, conciseness and flow. Grammatical correction is highly recommended in this article, considering the international scope of journal.
Line 17 - citation – 12, 15, 16
Typo and grammatical errors in line 35. “Any “ instead of “ant”?
“Although FGM/C is a human rights violation, in 2016, it was estimated that, at least 200 49 million females in thirty countries were victimized by FGM/C [4].” The number seems too high. Also, the link in the reference is not working. Please ensure correct referencing and correct statistics.
Line 65 - author’s name is needed for 12, 13 citation.
Line 77, 78 - unnecessary capitalization of first letter, lack of semicolon and space before in-text citation.
Line 81 – Don’t start with “which”.
Line 87: Clarify what you write. Put numbers for all 5 steps of decision making process.
Line 93: A semi-colon is not needed, breaks the statement.
Rectify overall grammatical errors. For example: “No systematic review was carried on the FGM/C decision-making process DMP or 101 household decision-makers HHDM(s).” should be worded as “Previous systematic reviews have not addressed….” Or something like that.
At line 105, before moving to methods, write a clear objective at the end of background such as “The objective of this study is to .. … …”
Methods:
Line 138: What search limites did you implement in precise? You write a general statement but what terminology was used in the systematic review?
There is no information on quality and risk of bias of the studies. All your selected studies need to be graded for the quality of evidence they provide. Based on this evidence, you make a decision about the internal and external validity of the findings. I did not find any paragraph on this. Please refer to https://ktdrr.org/products/update/v1n5/dijkers_grade_ktupdatev1n5.pdf - I recommend adding a paragraph and a table grading all your included studies.
There is clearly no way a reader can figure out the analytical part of the study. How was the individual data from each study synthesized? How were the studies categorized in terms of outcomes?
For qualitative analysis, you mention narrative synthesis, which is a very generic term. What precise method did you use to identify the themes and other qualititative data from the studies? There is no mention of this. Did you use any software such as nVivo? OR how do you grade the reliability of the information you identified from the study. This is unclear and needs a major explanation.
Results:
Please work on the writing. You clearly mean countries but have mistyped counties which can mean a major misinterpretation, especially when the journal has an international outreach. I emphasize again to work with a scientific editing team. What is FGD? Is it focus group discussion? You never explain the abbreviation anywhere in the paper. Things like these are basics to be considered before submitting a manuscript.
Discussion looks good, but are totally contingent upon better narration of background, methods and results.
Author Response
Good focus on an understudied public health issue. However entire article needs a proof reading my a scientific writing expert. There are a lot of grammatical errors and writing issues in terms of conciseness, clarity and flow. Please consult a writing expert and get the entire manuscript corrected.
Abstract – FGM/C abbreviation needs to be explained at first place. What is /C? Poorly worded abstract. Please rectify all grammatical errors.
Explanation of FGM/C abbreviation was added to the abstract “Female genital mutilation/cutting”. Grammatical errors were reviewed.
Background:
The background needs a lot of work in terms of clarity, conciseness and flow. Grammatical correction is highly recommended in this article, considering the international scope of journal.
Line 17 - citation – 12, 15, 16
Done
Typo and grammatical errors in line 35. “Any “ instead of “ant”?
Done.
“Although FGM/C is a human rights violation, in 2016, it was estimated that, at least 200 49 million females in thirty countries were victimized by FGM/C [4].” The number seems too high. Also, the link in the reference is not working. Please ensure correct referencing and correct statistics.
“200 million females in thirty country were victimized by the practice” the statement was verified from the reference; and the link in the reference was corrected to: <https://www.unicef.org/media/files/FGMC_2016_brochure_final_UNICEF_SPREAD.pdf> additionally the information was checked and other source of the information was found at <https://www.who.int/reproductivehealth/topics/fgm/prevalence/en/> .
Line 65 - author’s name is needed for 12, 13 citation.
Done.
Line 77, 78 - unnecessary capitalization of first letter, lack of semicolon and space before in-text citation.
Fixed on line 77,78
Line 81 – Don’t start with “which”.
The word “Which” was replaced with “this”
Line 87: Clarify what you write. Put numbers for all 5 steps of decision making process.
Steps were numbered.
Line 93: A semi-colon is not needed, breaks the statement.
The semi-colon was deleted.
Rectify overall grammatical errors. For example: “No systematic review was carried on the FGM/C decision-making process DMP or 101 household decision-makers HHDM(s).” should be worded as “Previous systematic reviews have not addressed….” Or something like that.
Sentence was corrected.
At line 105, before moving to methods, write a clear objective at the end of background such as “The objective of this study is to .. … …”
Aim of the study was stated.
Methods:
Line 138: What search limites did you implement in precise? You write a general statement but what terminology was used in the systematic review?
Search terms and limits to the search were mentioned
There is no information on quality and risk of bias of the studies. All your selected studies need to be graded for the quality of evidence they provide. Based on this evidence, you make a decision about the internal and external validity of the findings. I did not find any paragraph on this. Please refer to https://ktdrr.org/products/update/v1n5/dijkers_grade_ktupdatev1n5.pdf - I recommend adding a paragraph and a table grading all your included studies.
Critical and ethical appraisal originally was part of the manuscript but was removed before submission to the journal, upon your guidance it was added inform of text and tables to the manuscript, refer to table 3 and 4. The tables contain the +/- grading system, while text explain why each study was removed after the critical appraisal.
There is clearly no way a reader can figure out the analytical part of the study. How was the individual data from each study synthesized? How were the studies categorized in terms of outcomes?
A table is included
For qualitative analysis, you mention narrative synthesis, which is a very generic term. What precise method did you use to identify the themes and other qualititative data from the studies? There is no mention of this. Did you use any software such as nVivo? OR how do you grade the reliability of the information you identified from the study. This is unclear and needs a major explanation.
Quality appraisal table is included, No software was use to analyse the findings.
Results:
Please work on the writing. You clearly mean countries but have mistyped counties which can mean a major misinterpretation, especially when the journal has an international outreach. I emphasize again to work with a scientific editing team.
Counties was replaced with countries
What is FGD? Is it focus group discussion? You never explain the abbreviation anywhere in the paper. Things like these are basics to be considered before submitting a manuscript.
Abbreviation explanation was added to table 1.
Discussion looks good, but are totally contingent upon better narration of background, methods and results.
Reviewer 2 Report
Authors present a systematic review about decision making process in female genital mutilation. Study contains very complex presentation of used material and methods. I would suggest to include some information about how female genital mutilation affects women from psychological point of view. Manuscript is written clear and do not contain language errors. Analyzed study material is broad. Presented table contains main important information from each study that are used to create this systematic review. In conclusion I think this subject was presented widely and profound.
Author Response
Authors present a systematic review about decision making process in female genital mutilation. Study contains very complex presentation of used material and methods. I would suggest to include some information about how female genital mutilation affects women from psychological point of view. Manuscript is written clear and do not contain language errors. Analyzed study material is broad. Presented table contains main important information from each study that are used to create this systematic review. In conclusion I think this subject was presented widely and profound.
The psychological impact was originally mentioned briefly in the introduction part of the research as a consequence of FGM/C practice. More information regarding FGM/C psychological impact was added to introduction part, using a reference that was already used in the introduction- thus no new reference number or citation was added.
Reviewer 3 Report
First of all,this is an interesting approach.
Nevertheless, the manuscript has to be improved significantly. Hypotheses and conclusions are mixed up.
A systematic outline without repetitions - sometimes repeated generalities- is necessary. A suggestion for this outline is
1. aim/special questions:
-> formulate the questions exactly, "decision-making process" and "The role of fathers" are no questions and very general
2. introduction
3. material and methods
some parts of "material and methods" are in the resulst and discussion part - this is not the right place
4. results/discussion/ special questions/maybe include limitations:
compare the studies under your questions and answer your questions in a readable way. To refer to the somalia study and repeat the table is no discussion.
As a reader, I want to get an overview in a comprehensive and open way a synthesis of your research.
Recommendation: For identification of studies, list author, year, not "Somalia" eg: (Akinsulure-Smith, 2017)
5. conclusion
Sorry, but this part is full of generalities, hypotheses and wishes. It should be rewritten.
I dont think it is a maladministration if it is not possible it make a worldwide conclusion. Try to sum up, what is generalizable of your research and what is not generalizable. Explain why it is not generalizable in the discussion part.
In detail:
line 11 insert (FGM/C) abrevaiation should be explained as they appear
line 34 shorten and focus, maybe replace "endless" with lifelong
line 36 complications.. maybe insert brackets (severe pains, wound healing disorders, bleeding to death)
explanation: All unwanted consequences are complications, regardless of severity of consequences
line 39 what is neccessary suffering? please delete or explain
line 42 incorrect citation, type II includes partial or total excision of the labia minora
explanation: Type I already may include excision of the clitoris, necessary condition for type II is both, clitoris and labia minora partially or total
line 46 Type IV please cite correct: "unclassified...that...
explanation: The WHO definition doesn`t make differences concerning the intention of FGM, it refers to affected anatomic structures
line 62 list the studies first by author an year
e.g: xy (2011), ab (2012) and cv (2013) reported ...
line 78- 81: "Social norms...society". add document/citation - otherwise it is a personal statement. Sum up refering to the reference.
87 - 92: sentence over several lines, cluttered
line 94: see above as explained for line 62, which studies?
line 102 - 15 delete sentence, focus, decribe and explain aim of the study
line 109 - 111 explain shortly, how you checked that your aim was not already fulfilled by other research
line 140 - 141 which reference list
line 142 - 143 delete sentence, what you did not do, you dont have to mention
line 168 - 170 belongs to part limitations
line 174 - 193 delete all repetitions, (what is already in the table) avoid repetitions
proposal for part
"Study selection":
delete line 153- 154.
Inclusion and exclusion criteria are listed in table 2.
....table 2...
Then describe everything what is important to know and not mentioned in the table. If you explain your stages, refer with numbers to your table. Explanation: by redaing the text, the reader does not know, what and where is the "third stage" (line 184)
line 195 delete the second part of the sentence
line 202 delete sentence The...
line 205 Mention first organizing and analyzing the data, then narrative synthesis, delete repetion in line 2028 - 209
line 251- 258 no information, the reader has to study the table. Please rewrite.
Table 3 has to be rewritten.
For example in the part aim, dont make long sentences, if there is more than one aim, make a list
Avoid "the majority", "many", list the percentages.
And, what are "young parents"? Between 16 and 26 years old? Between 20 and 30 years?
These are only some examples, the table has to be carefully rewritten in a way that the reader knows what exactly was explored and what wre the results.
line 273 - 275 Is this a hypothesis, an interpretation? If it is a conclusion, it should be exactly proved why.
The discussion part should be rewritten. Avoid generalities (line 390 -391). i dont understand, what line 392 - 401 have to do with the study.
line 402 "some communities"? which communities? This is a sentence over 5 lines. Please clarify.
line 444 - 454 Sum up the role of men in an understandable way. How influence the men the decision making process? In which communities, ethnicitis?
line 455 - 458 I dont understand. Please clarify.
Strength and limitations has to be rewritten without repetions that were mentioned before. Compare the results in a greater context, this is also necessary for the discussion part.
line 475 - 489 conclusions, see above
Author Response
First of all,this is an interesting approach.
Nevertheless, the manuscript has to be improved significantly. Hypotheses and conclusions are mixed up.
A systematic outline without repetitions - sometimes repeated generalities- is necessary. A suggestion for this outline is
- aim/special questions:
-> formulate the questions exactly, "decision-making process" and "The role of fathers" are no questions and very general
A research question is added.
- introduction
- material and methods
some parts of "material and methods" are in the resulst and discussion part - this is not the right place
Corrected
- results/discussion/ special questions/maybe include limitations:
compare the studies under your questions and answer your questions in a readable way. To refer to the somalia study and repeat the table is no discussion.
As a reader, I want to get an overview in a comprehensive and open way a synthesis of your research.
Recommendation: For identification of studies, list author, year, not "Somalia" eg: (Akinsulure-Smith, 2017)
We mentioned it by country because the practice differ from country to another and if we mentioned it by author and year the text will be long
- conclusion
Sorry, but this part is full of generalities, hypotheses and wishes. It should be rewritten.
I dont think it is a maladministration if it is not possible it make a worldwide conclusion. Try to sum up, what is generalizable of your research and what is not generalizable. Explain why it is not generalizable in the discussion part.
Done
In detail:
line 11 insert (FGM/C) abrevaiation should be explained as they appear
Abbreviation was inserted.
line 34 shorten and focus, maybe replace "endless" with lifelong
replaced, and the sentence was divided to two sentences
line 36 complications.. maybe insert brackets (severe pains, wound healing disorders, bleeding to death)
Brackets were inserted for FGM/C complications
explanation: All unwanted consequences are complications, regardless of severity of consequences
line 39 what is neccessary suffering? please delete or explain
The unnecessary suffering and health inequality were explained by that it has no health benefits for women who are exposed to the practice, and causing them complications that can last a lifetime.
line 42 incorrect citation, type II includes partial or total excision of the labia minora
explanation: Type I already may include excision of the clitoris, necessary condition for type II is both, clitoris and labia minora partially or total
fixed
line 46 Type IV please cite correct: "unclassified...that...
Type IV includes any other procedures that fall under the WHO definition of FGM/C “all procedures …. or other injury to the female genital organs for non-medical reasons”, which mention the intention in “non-medical reason”. Type IV was rephrased in summarized manner and the intention phrase was removed.
explanation: The WHO definition doesn`t make differences concerning the intention of FGM, it refers to affected anatomic structures
line 62 list the studies first by author an year
e.g: xy (2011), ab (2012) and cv (2013) reported ...
Done
line 78- 81: "Social norms...society". add document/citation - otherwise it is a personal statement. Sum up refering to the reference.
It is a conclusion of the mentioned above, there is no reference
87 - 92: sentence over several lines, cluttered
I tried to summarize the sentence without affecting the meaning
line 94: see above as explained for line 62, which studies? Studies added
line 102 - 15 delete sentence, focus, decribe and explain aim of the study
sentence deleted; aim added
line 109 - 111 explain shortly, how you checked that your aim was not already fulfilled by other research
It is explained in that paragraph
line 140 - 141 which reference list
Reference harvesting system was used
line 142 - 143 delete sentence, what you did not do, you dont have to mention
done.
line 168 - 170 belongs to part limitations
deleted as already mentioned in the Limitation part of the research
line 174 - 193 delete all repetitions, (what is already in the table) avoid repetitions
proposal for part
"Study selection":
delete line 153- 154.
Inclusion and exclusion criteria are listed in table 2.
Done
....table 2...
Then describe everything what is important to know and not mentioned in the table. If you explain your stages, refer with numbers to your table. Explanation: by redaing the text, the reader does not know, what and where is the "third stage" (line 184)
line 195 delete the second part of the sentence
Done
line 202 delete sentence The...
Done
line 205 Mention first organizing and analyzing the data, then narrative synthesis, delete repetion in line 2028 – 209
done.
line 251- 258 no information, the reader has to study the table. Please rewrite.
Table 3 has to be rewritten.
For example in the part aim, dont make long sentences, if there is more than one aim, make a list
Avoid "the majority", "many", list the percentages.
And, what are "young parents"? Between 16 and 26 years old? Between 20 and 30 years?
These are only some examples, the table has to be carefully rewritten in a way that the reader knows what exactly was explored and what wre the results.
After adding critical appraisal tables (table 3,4), The data extraction table was numbered (table 5). The data extraction table was re-written, after returning to the articles included in the table. For two articles “Most and many” words were used in the table as both articles generated qualitative results, and those words were used by the articles’ authors to reflect the findings without the use of any percentage or numbers. Age groups and gender of the participants were added if it was mentioned in the articles and had reflective power on the result. Summarizing and rephrasing of some information was made. Reparative information and conclusive statement were removed from the “Result column” in the table.
The studies in the results section is refer to by location and “author, year” because the location of the study is important to reflect how FGM/C is practiced differently according to location.
line 273 - 275 Is this a hypothesis, an interpretation? If it is a conclusion, it should be exactly proved why.
The discussion part should be rewritten. Avoid generalities (line 390 -391). i dont understand, what line 392 - 401 have to do with the study.
In this part of the discussion, the author is explaining the shift that took place in DMP, and how communities are now having to revaluate their decision regarding the practice after being presented with new information due to anti-FGM/C campaign for example.
line 402 "some communities"? which communities? This is a sentence over 5 lines. Please clarify.
Done. Modified.
line 444 - 454 Sum up the role of men in an understandable way. How influence the men the decision making process? In which communities, ethnicitis?
Done. Modified.
line 455 - 458 I dont understand. Please clarify.
Done. Modified.
Strength and limitations has to be rewritten without repetions that were mentioned before. Compare the results in a greater context, this is also necessary for the discussion part.
Done. Modified.
line 475 - 489 conclusions, see above
Done
Round 2
Reviewer 1 Report
The revised version is acceptable for publication. Thank you for the revisions.
Author Response
Done. See manuscript
Reviewer 3 Report
I think the manuscript has been significantly improved.
Nevertheless, a careful revision has to be done.
First, times are mixed up. I think what you did, should be written consistently in the past. ( e.g. see line 13 vs. line 16 ).
in detail:
line 22 Is this a hypothesis or a wish? If it is a conclusion of your study, then (maybe) write: "It was shown... that open the dialogue...leads to..."
If it is a wish or a hypothesis, I think "may" instead of "can" is more precise.
This applies also for the next sentence. Maybe you can precise what you found about the role of the fathers already in the abstract.
line 33 - 38 always "it can cause". Maybe: "FGM/C doesnt have any health benefits. It can causes .... sum up complications, including health inequality. Avoid repetitions.
I prefer to delete "unnecessary suffering" and maybe use the word "suffering". Unnecessary suffering is very general and it prepurposes that there is such a thing like "necessary suffering". Then you have to explain what is necessary suffering.
line 41 proposal: "FGM/C is classified into four types...[reference]"
line 56, 57 grammatical time and grammatical person correct?
line 62 Almroth.... Kokoui et al (2017) reported from studies in Somalia, ...and... that FGM/C...
This makes the text longer, but if you want to get the information, it should be mentioned as you did in the original manuscript.
line 76 grammatically correct? maybe "This practice...
line 80 I think not "Which" but "This". Make two sentences. Or don't use a semikolon. But then it is a long sentence.
line 84 participate not participant?
line 93 delete comma after Bjälklander
line 103 There is no verb in the sentence
line 117 grammatically correct? to searching? to search? for searching?
table 1 FGD/s should be in brackets
line 131 easier to read is: The following keywords were used ... : father...
I dont know, wether the keywords were used alone ( father "or" mother) or "in combination". There is written both.
line 138 "the English language"? correct?
line 139 delete semikolon
line 142 delete "Prisma flowchart" Figuer 1 in brackets is enough.
Prisma Flowchart
Eligilibility "with reasons" which reasons? short summary of the reasons is needed is this line 168 - 171? So "DMP and decision makers were not the primary aim of the study"? Please clarify.
line 178 see comment
line 151 To avoid duplication bias, before implementation of....removal...was done ..by ref works ... and manually
line 155 time ?
line 157 - 159 "to scanning"?
very long sentence,
Correct time?
"in which they were not indexed" - what do you mean?
line 160 time?
line 178 see comment to line 142, (Figure 1) is enough.
line 212 Write: Six studies were excluded according to the following reasons: one study...
Otherwise one does not understand the connection.
line 217 "(Table 3 and 4)" is enough.
line 232 "in the African countries or counties or both" ?
maybe "in Africa ( Ethiopia, ...)"
mideastern country iraq, kurdistan region? in the kurdistan region of iraq? please clarify
line 235 comma or "and", not both
line 236 delete "while", otherwise it is hard to understand
line 247
I recommend to write it as following
"chracteristics, design ... and summary are prestented in table 5. The line 247 - 250 delet refere to table 5
delete "of the 17 studies". It is already written in line 231
Table 5
Sometimes you wrote "sample". I am not sure if you mean the authors or the people studied. I think, if you mean the people studied, it is better to call them as humans.
line 259 -263 this is not a useful information for the reader. Which synthesis? What are you talking about? Delete
line 269 - 271 reference missing. Where does this statement come from?
line 271 replace "another study" with "xy (year)
line 274 list the authors and year
What does it mean "when it comes to the decision making process"?
line 276 verb is missing
line 278 - 280 reference is missing
line 280- 282 I don't think the study participants shared similar findings.... either the study revealed similar findings or the study participants reported simimilar answers...
line 282 delete "in", delete the comma maybe "Keita and Blankhard (2012, Guinea) reoported about malinke families ...[refernce number].
line 284 delete whereas, delete an or the comma,
line 289 delete "nevertheless"
line 293 delete "this paper reveals the difference". Describe what you found, for example: comparing the DMP at the Iraqi Kurdistan region with the the DMP in African countries...
part "Are females the decision-makers?"
As mentioned before, the authors should be named, not only "a study"
Mabe you write "These findings were supported by xy (study in Somaliland) [reference]. Mothers ...
or "ab ( Ethiopia) comes to the same conclusion [reference]."
I am not sure if "a study can share similar results". Ask an English native speaker please.
Part "The fathers role as decision makers"
see Part "Are females...
Please list the authors so that the reader can follow.
Note that as a reader I would like to know clear sentences what the results of the study were.
If i want more information, I can have a look in your table. If I need additional information, I have to read the article(s).
Delete unneccessary words like "alternatively", "whereas" because this causes confusion. You may use "on the other hand" if you decribe an opposite/contradictionary finding rs. "similarly" if you want to describe a match.
line 356 delete "The paper revealed" because you are talking about practise in the Bale Zone, Ethiopia.
Discussion part
line 418 delete "whereas", sentence is to long. (5 lines)
Conclusions
delete the first sentence. I would write: For the first time, this review provides summarized informations about DMP and HHDM.
line 442 delete "from the data of this review. You are the whole time talking abut this data.
line 443 introduces(!) the presence...
line 444 Proposal: ....the presence of silence in discussing FGM/C caused by social and cultural obligation influences DMP.
and cause confusion - what causes confusion? Make two sentences.
line 447 Proposal: It was shown, that DMP is important in FGM/C in different countries and ethnies. Due to limited studies, further research is needed.
line 447, at the end: "The need of an open dialogue/ opening the dialogue between males and females was shown because this can lead/ may lead (?) to a productive DMP." sounds better for me.
Then start a new sentence...
line 452 - 455 may or can?
Great work with minor revisions needed!
Author Response
Please the attachment